# SIG-1451, a Novel, Non-Steroidal Anti-Inflammatory Compound, Attenuates Light-Induced Photoreceptor Degeneration by Affecting the Inflammatory Process

**DOI:** 10.3390/ijms23158802

**Published:** 2022-08-08

**Authors:** Yuki Kikuchi, Eriko Sugano, Shiori Yuki, Kitako Tabata, Yuka Endo, Yuya Takita, Reina Onoguchi, Taku Ozaki, Tomokazu Fukuda, Yoshihiro Takai, Takahiro Kurose, Koichi Tanaka, Yoichi Honma, Eduardo Perez, Maxwell Stock, José R. Fernández, Masanori Tamura, Michael Voronkov, Jeffry B. Stock, Hiroshi Tomita

**Affiliations:** 1Laboratory of Visual Neuroscience, Graduate Course in Biological Sciences, Iwate University Division of Science and Engineering, 4-3-5 Ueda, Morioka 020-8551, Iwate, Japan; 2Rohto Pharmaceutical Co., Ltd., 6-5-4 Kunimidai, Kizugawa 619-0216, Kyoto, Japan; 3Signum Biosciences, 4999 Pearl East Circle, Boulder, CO 80301, USA; 4Department of Molecular Biology, Princeton University, Princeton, NJ 08544-1014, USA

**Keywords:** age-related macular degeneration, light-induced photoreceptor degeneration, anti-inflammatory drug, Toll-like receptor 4

## Abstract

Age-related macular degeneration is a progressive retinal disease that is associated with factors such as oxidative stress and inflammation. In this study, we evaluated the protective effects of SIG-1451, a non-steroidal anti-inflammatory compound developed for treating atopic dermatitis and known to inhibit Toll-like receptor 4, in light-induced photoreceptor degeneration. SIG-1451 was intraperitoneally injected into rats once per day before exposure to 1000 lx light for 24 h; one day later, optical coherence tomography showed a decrease in retinal thickness, and electroretinogram (ERG) amplitude was also found to have decreased 3 d after light exposure. Moreover, SIG-1451 partially protected against this decrease in retinal thickness and increase in ERG amplitude. One day after light exposure, upregulation of inflammatory response-related genes was observed, and SIG-1451 was found to inhibit this upregulation. Iba-1, a microglial marker, was suppressed in SIG-1451-injected rats. To investigate the molecular mechanism underlying these effects, we used lipopolysaccharide (LPS)-stimulated rat immortalised Müller cells. The upregulation of C-C motif chemokine 2 by LPS stimulation was significantly inhibited by SIG-1451 treatment, and Western blot analysis revealed a decrease in phosphorylated I-κB levels. These results indicate that SIG-1451 indirectly protects photoreceptor cells by attenuating light damage progression, by affecting the inflammatory responses.

## 1. Introduction

Photoreceptor degenerative diseases such as retinitis pigmentosa and age-related macular degeneration (AMD) are major causes of blindness worldwide; moreover, the number of patients is only expected to increase with the continuing increase in the ageing population [1,2]. AMD is classified into two types: wet and dry AMD [3]. In wet AMD, the newly formed choroidal vessels (choroidal neovascularization; CNV) appear in the macular area, which causes the loss of visual function. The introduction of anti-vascular endothelial growth factor therapies is leading to meaningful reductions in patients with wet AMD. Various factors, such as the accumulation of unprocessed cellular debris [4,5,6] and increased oxidative stress [7,8,9], caused by dysfunction of retinal pigment epithelium (RPE) cells, are implicated in the development of dry AMD [10], for which no effective treatment has yet been developed.

Various animal models, such as chemical-induced photoreceptor degeneration [11,12], genetic models [13,14,15], and continuous light damage models [16,17], have been used to investigate the mechanisms of photoreceptor degeneration. Among these, light damage caused by continuous light exposure is widely used as a dry AMD model [18,19]. Maeda et al. [20] reported that excessive production of all-trans retinal due to continuous light exposure was involved in photoreceptor degeneration, as increased all-trans retinal induced rapid NADPH oxidase-mediated overproduction of intracellular reactive oxygen species [21]. In addition, excessive inflammation resulting from RPE cell dysfunction has also been associated with the progression of dry AMD [22].

Retinal inflammation is mainly initiated by Müller cells and microglia [22,23]. Recently, some reports have shown that excessive inflammation accelerates retinal degeneration [24,25,26]. Toll-like receptor 4 (TLR4) is well-known as one of the key receptors involved in the inflammatory response—it activates nuclear factor kappa B (NF-kB), leading to the upregulation of downstream genes, including several chemokines and cytokines. TLR4 upregulation was also observed in a light damage model that used Abca4-/- Rdh8-/- double knockout mice [27]. Therefore, it is believed that TLR4 plays an important role in AMD progression and that TLR4-related pathways may emerge as potential therapeutic targets for inhibiting excessive inflammation and regulating AMD progression.

Chronic atopic dermatitis is caused by the release of cytokines such as TNF-α and IL-6 [28]. SIG-1451, a novel non-steroidal anti-inflammatory drug candidate being developed for the treatment of atopic dermatitis, was found to inhibit lipopolysaccharide (LPS)-induced IL-6 production. AMD is also caused by the release of cytokines mediated by TLR4 [29]. Therefore, we hypothesised that SIG-1451 would protect photoreceptor cells from excessive inflammation. In the present study, we used a light-induced photoreceptor degeneration model to investigate the effects of SIG-1451 on photoreceptor cell death, as well as the molecular mechanisms of the protective effects, using immortalised rat retinal Müller cells (rMC-1 cells) stimulated with LPS, a TLR4 agonist, as an in vitro model of inflammation.

## 2. Results

### 2.1. SIG-1451 Rescues Retinal Thickness and Improves Electroretinographic Responses in Light-Induced Photoreceptor Degeneration

To evaluate the effect of SIG-1451 on light-induced photoreceptor degeneration, rat eyes were exposed to 3000 lx light for 24 h as a preliminary experiment. Intraperitoneal injections of 10 mg/kg or 100 mg/kg SIG-1451 in rats were not observed to have any significant protective effects on rat eyes exposed to 3000 lx of light, and electroretinogram (ERG) recordings did not indicate any protective effects either (Appendix A). However, SIG-1451-treated rats had increased retinal thickness and ERG amplitudes compared to those of control rats, indicating the tendency of SIG-1451 to produce some protective effects. Therefore, we used 1000 lx of light as the light damage condition to evaluate the protective effects of SIG-1451. Optical coherence tomography (OCT) images showed that the retinal layer (i.e., the inner limiting membrane-PE) and outer nuclear layer (ONL) in the superior part of the retina of SIG-1451-treated rats were thicker than those of saline-treated rats (Figure 1A). In contrast, in the inferior part of the retina, the whole retinal thickness of SIG-1451-treated rats was significantly thinner than that of saline-treated rats (Figure 1B). The ONL thickness in the superior and inferior parts of the retina in the saline group was 31.19 ± 1.80 and 38.33 ± 3.37 µm (mean ± SE), and that in the SIG-1451 group was 40.83 ± 2.01 µm and 39.59 ± 3.09, respectively. Thus, the superior part of the retina of saline-administered rats was highly affected by light damage. On the other hand, there was no significant difference in ONL thickness in the SIG-1451-group in either the superior or the inferior part of the retina.

Retinal function was evaluated based on ERG 3 d after light damage. Typical waveforms are shown in Figure 1C. The amplitudes of a- and b-waves in ERGs from SIG-1451 administered rats were significantly larger than those from saline-administered rats (Figure 1D,E).

### 2.2. SIG-1451 Protects the Photoreceptor Cells from the Light-Induced Photoreceptor Degeneration

Paraffin-embedded sections prepared from rat eyes 8 d after light damage were stained with haematoxylin-eosin for retinal thickness evaluation. Severe photoreceptor degeneration was observed in the superior part of retinas treated with saline as well as SIG-1451 (Figure 2A). However, significant protective effects of SIG-1451 were detected near the optic nerve (Figure 2B,C). We also observed the protective effects of SIG-1451 on ONL thickness in the inferior part of the retina (Figure 2C).

### 2.3. Decreased Iba-1-like Immunoreactivities in SIG-1451-Treated Retinas

Microglial activation and retinal layer microglial localisation in light-damaged retinas were investigated using an Iba-1 antibody. Iba-1-like immunoreactivity (Iba-1 IR) was negligible in retinas without light damage (Figure 3A). In saline-treated light-damaged retinas, increased Iba-1 IR was observed in the retinal layer (Figure 3B,C); however, this increase was weaker in the inferior part of the retina than that in the superior part. Moreover, Iba-1 IR was lower in all retinal layers of SIG-1451-treated rats compared to that in saline-treated rats, with a significant reduction in Iba-1 IR in the ganglion cell-inner plexiform layer of the superior part and the ONL of the inferior part of the retina. There was no difference in GFAP-IR between saline- and SIG-1451-treated retinas (Figure 3A). However, the GFAP-IR tended to increase compared to retinas without light damage.

### 2.4. Upregulation of CCR2 after the Photoreceptor Degeneration Was Inhibited by SIG-1451

TLR4, a key receptor involved in inflammatory responses, was found to be overexpressed in the retina 1 d after the continuous light exposure; moreover, there was no significant difference in its expression level between saline- and SIG-1451-treated rats (Figure 4A). C-C motif chemokine 2 (CCL2) expression was also significantly increased by light damage, with no observed difference between saline- and SIG-1451 treated rats (Figure 4B). The expression of complement component 3 tended to increase (Figure 4C), while that of C-C chemokine receptor type 2 (CCR2), known to be a receptor for CCL2, was dramatically suppressed compared to that in saline-treated rats (Figure 4D). GFAP expression was also suppressed by treatment with SIG-1451 (Figure 4E).

### 2.5. RT-qPCR Analysis of Inflammatory Response-Related Genes in Cultured rMC-1 Cells

To explore the possible mechanisms underlying the observed effects of SIG-1451, we used LPS-stimulated rMC-1 cells and investigated the expression of genes related to the inflammatory response. The concentration of SIG-1451 that was not toxic to cultured rMC-1 cells was used (Appendix A). TLR4 expression remained unchanged 6 h after LPS stimulation (Figure 5A) and tended to decrease due to SIG-1451 treatment at 24 h after LPS stimulation (Figure 5B). CCL2 expression was found to be increased markedly 6 h after LPS stimulation, and SIG-1451 treatment inhibited this increase (Figure 5C). At 24 h after LPS stimulation, CCL2 expression levels were almost the same in untreated cells and SIG-1451-treated cells, but there was a small but significant increase in LPS-stimulated cells treated with SIG-1451 (Figure 5D). C3 expression in LPS-stimulated cells increased significantly; moreover, this increase tended to be inhibited by SIG-1451-treatment (Figure 5E,F).

### 2.6. Western Blot Analysis of TLR4, NF-κB, I κB, and MAPK Expression

Western blot analysis showed that TLR4 levels in cells remained unchanged irrespective of the treatment (Figure 6A). It is known that the phosphorylation of NF-κB and I-κB plays central roles in immune system functioning. NF-κB exists in the cytoplasm in an inactive form when bound to I-κB. Various stimuli, such as LPS, inflammatory cytokines, and viruses, trigger the activation of I-κB kinases, which induce the phosphorylation of I-κB. NF-κB translocates into the nucleus because NF-κB is not able to bind to the phosphorylated- I-κB. Therefore, we investigated the levels of the phosphorylated forms of NF-κB and I-κB (pNF-κB and pI-κB, respectively). The level of pNF-κB was not different (Figure 6B), but the level of pI-κB was 2 times higher, and this increase was inhibited to a normal level by the addition of SIG-1451 (Figure 6B). The levels of pp38 (Figure 6C) and pJNK (Figure 6D) tended to increase, whereas that of pJNK2 increased significantly.

## 3. Discussion

The present study sought to determine whether anti-inflammatory drugs could have some protective effect on light-induced photoreceptor degeneration. Our in vivo experimental data indicated that the anti-inflammatory drug SIG-1451 did affect the pathways involved in light-induced photoreceptor degeneration and inhibited its progression partially. Taken together with the results of our in vitro experiments, this suggests that SIG-1451 did not act in the early phase of photoreceptor cell death, such as during an apoptotic process, and that SIG-1451 attenuated microglial activation by inhibiting the phosphorylation of I-κB, resulting in attenuation of the progression of degeneration.

Photoreceptor degeneration due to light damage is a well-established model for drug screening and exploring the mechanisms of photoreceptor cell death [1]. It is known that photoreceptor cell death due to light damage is characterized by more severe degeneration of the superior part of the retina than of the inferior part. We observed the same phenomena based on OCT imaging (Figure 1A,B) and morphological evaluation (Figure 2A,B). The protective effects of SIG-1451 observable on OCT images obtained 1 d after light damage were observed in the superior part of the retina (Figure 1A) but not in the inferior part. The role of microglia in photoreceptor degenerations is still controversial. Funatsu J et al. [30] reported that the depletion of resident microglia exacerbated cone cell death in rd10 mice. On the other hand, some researchers have shown that the increase in inflammation mediated by activated microglia is responsible for the pathogenesis and progression of AMD [25,26]. Iba-1-like immunoreactivity markedly increased in the superior part of the retina; moreover, SIG-1451 inhibited this increase (Figure 3A). Therefore, a primary effect of SIG-1451 may be to inhibit secondary degeneration mediated by inflammation, although it does not appear to directly affect photoreceptor cell death, such as the apoptotic process.

Jiao et al. reported that photo-oxidative damage in the retina was attenuated by the depletion of C3, which is expressed in abundance in macrophages [31]. Genome-wide association studies have shown that disease progression in AMD is associated with complement factor H [32]. Subsequent investigations have revealed that other complement genes, such as C3 and C2, and complement factor B, are also risk factors for AMD [33]. Therefore, innate immunity has a strong association with AMD. In retinas exposed to light damage, CCL2 expression showed an increase of up to 250 times compared to that in controls (no LD, Figure 4B), and C3 expression tended to increase with or without SIG-1451 treatment. It is well-known that Müller cells protect neurons via a release of neurotrophic factors, the secretion of antioxidants, and the uptake or degradation of the excitotoxins by responding to various types of injury and accelerating neuronal degeneration through a reactive Müller cell gliosis [34]. To focus on the role of Müller cells in the protective effect of SIG-1451, we used the immortalized rat Müller cell line, rMC-1 cells. CCL2 and C3 expression in LPS-stimulated cultured rMC-1 cells increased up to approximately 40 times and 20 times, respectively, compared to that in controls (no LPS, Figure 5B,C). Furthermore, SIG-1451 blocked this increase. It has been suggested that various types of cells in the retina express CCL2 and C3, and SIG-1451 predominantly blocks their expression in Müller cells. In addition, CCR2 expression was not detected in cultured Müller cells, although its expression was markedly increased in the LD-retina, which was inhibited by SIG-1451 administration (Figure 4D). Ma W et al. reported that the gene expression of chemotactic cytokines and cell adhesion molecules in RPE cells was altered by co-culture of activated microglia and that mRNA expression of CCL2 in the RPE cells was significantly increased by co-culture with activated microglia [30]. Retinal microglia translocate into the outer retina [35] and accumulate in the subretinal space [36] under conditions of advanced age and photoreceptor injury. RPE cells may be involved in the inflammatory responses. CCR2 has been reported to be expressed in macrophages in an inherited retinal degeneration model [37]. These results indicate that SIG-1451 might inhibit CCR2 expression in other retinal cells, such as macrophages activated by LD and the RPE cells, but not in Müller cells in vivo.

GFAP expression increases in Müller cells in late AMD patients [38]; moreover, changes in GFAP expression occur in response to various types of retinal insults, such as laser- or light-induced damage, diabetic retinopathy [39,40,41], retinal detachment [42], and inherited retinal degeneration [43]. It is thought that increased GFAP expression in inherited retinal degeneration and AMD may reflect the secondary response to a primary insult to another cell type. SIG-1451 blocked GFAP and CCR2 expression at 6 h after light exposure (Figure 4D,E), and there was no difference in GFAP-IR 8 days after light exposure (Figure 3A), which indicated that SIG-1451 affected the early phase of the secondary response to light damage.

The expression of C-C chemokine family members, such as CCL2, is regulated by two different pathways—the NF-κB [44] and mTORC1-FOXK1 [45] pathways. In classical NF-kB-dependent pathways, activated I-kB kinases phosphorylate I-kB, which leads to I-kB ubiquitination and degradation, resulting in NF-κB dimer, p50, and p65 translocation into the nucleus [46]. Western blot analysis indicated that LPS stimulation increased the level of pI-kB, but not pNF-κB. We previously reported that bright light induced the translocation of p65 to the mitochondria from the cytosol immediately after light exposure [47]. Translocation of p65 and p50 from the cytosol to the mitochondria has also been reported in cytokine-stimulated cells [48]. Thus, the regulation of the C-C chemokine family might involve a similar pathway, even though the trigger in our case was bright light, not cytokines.

## 4. Materials and Methods

### 4.1. Animals

Sprague Dawley rats (CLEA Japan, Inc., Tokyo, Japan) were housed under 12 h dark and 12 h light (5 lx) conditions, with available chow and water. They were kept in the dark for 24 h, and the pupils were dilated with tropicamide (Midrin-P, Santen Co., Ltd., Osaka, Japan) before the induction of light damage exposure to 1000 lx white fluorescence in a light box (041001, NK system, Osaka, Japan) for 24 h. All the experimental schedules are shown in Appendix A. All animal care was strictly in conformance with the ARVO Statement for the Use of Animals in Ophthalmic and Vision Research and the Iwate University Guidelines for Animals in Research.

### 4.2. Optical Coherence Tomography (OCT)

OCT was performed 1 day after light damage induction. Rats were anaesthetised by intramuscular injection of 75 mg/kg ketamine and 0.5 mg/kg medetomidine, and their pupils were dilated using tropicamide. Image acquisition of a 1.2 mm length of the rat retina, including the optic disk, was performed using the line scan mode on an OCT imaging device equipped with a special ordered lens (RS-3000, NIDEK Co., Ltd., Aichi, Japan).

### 4.3. ERGs

ERGs were obtained 3 d after the end of light exposure with PuREC (Mayo Corporation, Aichi, Japan). The rats were dark-adapted overnight and then anaesthetised by intramuscular injection of 45 mg/kg ketamine and 4.5 mg/kg xylazine and topical eye anaesthesia (Benoxil ophthalmic solution 0.4%; Santen Co., Ltd., Osaka, Japan). The pupils were dilated using tropicamide, and a small contact lens with an electrode was mounted on the cornea with hydroxyethyl cellulose eye solution (Scopisol 15^®^; Senju Pharmaceutical Co., Ltd., Osaka, Japan). A reference electrode (crocodile-mouth electrode) was placed in the oral cavity. The high- and low-path filters were set to 0.3 Hz and 500 Hz, respectively.

### 4.4. Paraffin-Embedded Sections and HE Staining

Eight days after the end of light exposure, the rats were euthanised by CO_2_ inhalation. Eyes were enucleated, fixed, and embedded in paraffin. The blocks were sectioned (5 μm thick) along the vertical meridian to allow for the comparison of all regions of the retina in the superior and inferior hemispheres. The sections were stained with haematoxylin and eosin, and retinal thickness was measured at 500 µm intervals from the centre of the optic nerve head.

### 4.5. Iba-1 and GFAP Immunohistochemistry

Paraffin-embedded sections were deparaffinised according to standard procedures. The sections were then incubated in citrate buffer (pH 6.0) in a microwave oven for antigen retrieval. After blocking with 3% normal goat serum, sections were incubated in a solution of Iba-1 antibody (1/1000, FUJIFILM Wako Pure Chemical Corporation, Osaka, Japan) and GFAP antibody (1/50, Cell Signaling Technology, Tokyo, Japan) or in a control solution of normal rabbit IgG and normal mouse IgG overnight at 4 °C. After washing, sections were incubated with Alexa594-conjugated anti-rabbit IgG at room temperature for 30 min. After washing, the sections were covered with mounting media, including 4′,6-diamidino-2-phenylindole (DAPI; VECTASHIELD, Funakoshi, Tokyo). Iba-1 immunoreactivity data were obtained using a macro on an ImageJ.

### 4.6. RT-qPCR of Inflammatory Response-Related Genes

Total RNA from rat retina or cultured rMC-1 cells were extracted using the Absolutely RNA Miniprep Kit (Agilent Technologies, Tokyo, Japan) and ReliaPrep^TM^ RNA Cell Miniprep System (Promega, Tokyo, Japan), respectively. cDNA was synthesised using the ReverTra Ace^®^ qPCR RT Master Mix with gDNA remover (TOYPBO, Osaka, Japan). RT-qPCR was performed using the SsoAdvanced^TM^ Universal SYBR^®^ Green Supermix (Bio-Rad Laboratories, Tokyo, Japan). The primers used are listed in Table 1. RNA expression levels were quantified using the CFX Connect Real-Time PCR Analysis System (Bio-Rad Laboratories, Tokyo, Japan). GAPDH was used as the reference gene, and expression was quantified using the comparative Ct method.

### 4.7. Reagents

LPS (Sigma Aldrich, Tokyo, Japan) was dissolved in Dulbecco’s Modified Eagle’s Medium (DMEM) to prepare a 1 mg/mL solution. The stock solution was then diluted with an FBS-free medium to adjust the concentration to 1 µg/mL. SIG-1451 working solution was also prepared by dissolving in FBS-free DMEM.

### 4.8. Cell Culture

Immortalised rat rMC-1 cells, derived from the stable transformation of SV40 antigen into primary rat retinal Müller cells, obtained from Applied Biological Materials Inc. (T0576; Richmond, BC, Canada) [1] were cultured in DMEM containing 10% FBS (Thermo Fisher Scientific, Tokyo, Japan), 1% antibiotics (Thermo Fisher Scientific, Tokyo, Japan), 1% GlutaMax (Thermo Fisher Scientific, Tokyo, Japan), and 0.35 *w*/*v*% D-glucose (FUJIFILM Wako Pure Chemical Corporation, Osaka, Japan) at 37 °C in a 5% CO_2_ atmosphere on culture plates coated with atelocollagen (KOKEN Co., Ltd., Tokyo, Japan). Before LPS and/or SIG-1451 were added to the culture medium, cells were incubated in an FBS-free medium for 24 h. For RNAs and protein extraction, cells were collected 6 and 24 h after LPS and/or SIG-1451 addition.

### 4.9. Western Blotting

rMC-1 cells were seeded (1.5 × 10^6^ cells per well) in 6 cm plates. Western blot analysis was performed as previously described [2]. Briefly, total cell lysates were prepared using Pierce^®^ RIPA Buffer (Thermo Fisher Scientific, Tokyo, Japan) and Halt^TM^ Protease and Phosphatase Inhibitor Single-Use Cocktail, EDTA-free (100×; Thermo Fisher Scientific, Tokyo, Japan) and 0.5 M EDTA solution (Thermo Fisher Scientific, Tokyo, Japan). Protein concentration was measured using a BCA assay kit (Thermo Fisher Scientific, Tokyo, Japan). Thirty micrograms of protein per sample were loaded onto a 4–15% mini-protean TGX precast polyacrylamide gel (Bio-Rad Laboratories, Tokyo, Japan) and then transferred to a polyvinylidene fluoride membrane (Bio-Rad Laboratories, Tokyo, Japan). After blocking with Block Ace (KAC Co., Ltd., Kyoto, Japan), the membranes were incubated with primary antibodies listed in Table 2. After washing, the membrane was incubated with an alkaline phosphatase-conjugated secondary antibody. Chemiluminescence detection using CDP-star (GE Healthcare, Tokyo, Japan) was performed according to the manufacturer’s protocol. Band density was measured using ImageQuant (GE Healthcare, Tokyo, Japan).

### 4.10. Statistical Analysis

Tukey’s multiple comparison test and Student’s *t*-test were used for the comparative analyses. Statistical analyses of the in vitro experiments were performed using GraphPad Prism (MDF, Tokyo, Japan).

## Figures and Tables

**Figure 1 ijms-23-08802-f001:**
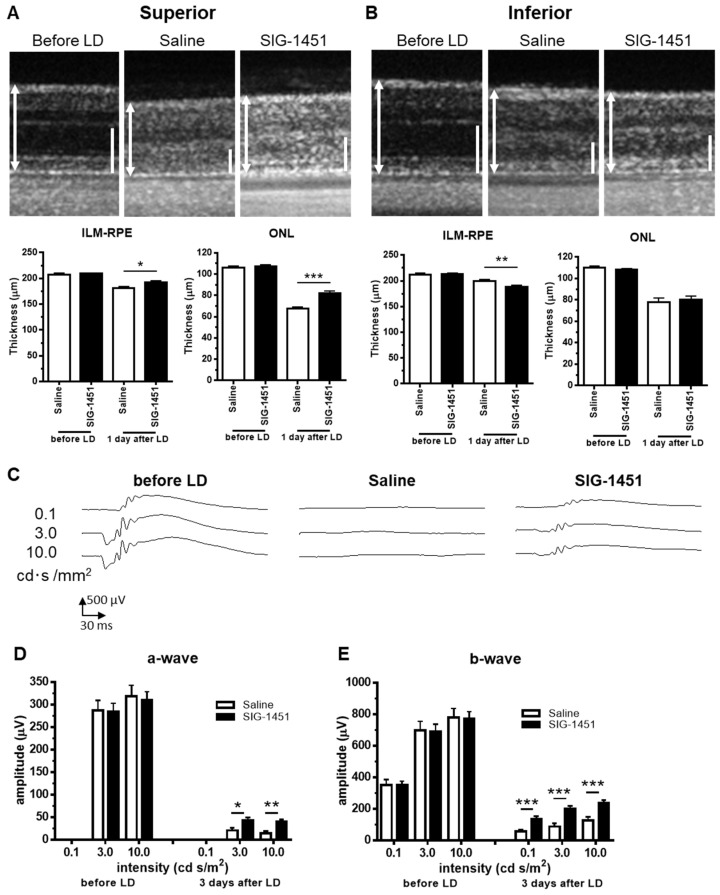
Effects of SIG-1451 on retinal thickness and electroretinographic responses in light-induced photoreceptor degeneration. OCT images of the superior (**A**) and inferior part (**B**) of retinas were obtained 1 d after the end of light exposure, and the thicknesses of the inner limiting membrane (ILM) to the retinal pigment epithelium (RPE) and the outer nuclear layer (ONL), including outer segments, were measured. ILM-RPE and the ONL are indicated by a double-headed arrow (L) and bar (R), respectively. Typical waveforms of ERGs are shown in (**C**). The amplitudes of a-waves (**D**) and b-waves (**E**) were measured. Data are represented in terms of the mean ± SEM values (saline: *n* = 14, SIG-1451: *n* = 16, unpaired *t*-test; *, **, *** *p* < 0.05, 0.01, 0.001).

**Figure 2 ijms-23-08802-f002:**
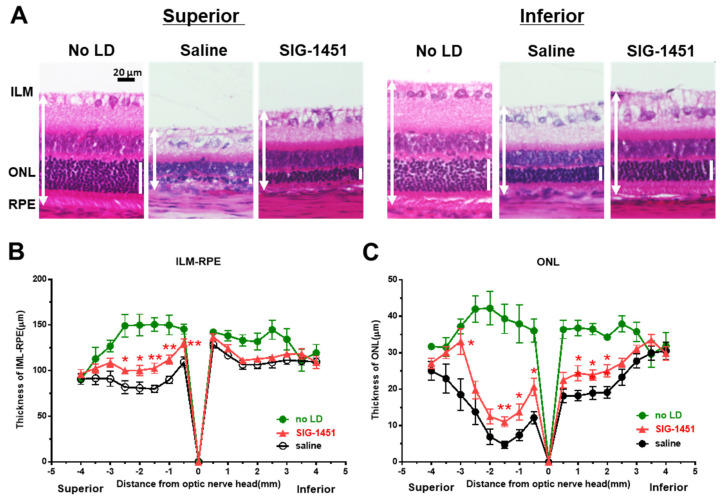
Histological examination of retinas 8 d after light damage. Typical microphotographs of the superior and inferior parts of the retina (**A**). ILM-RPE (**B**) and ONL (**C**) thicknesses in saline- and SIG−1451-treated rats. Data are represented in terms of the mean ± SE values (saline: *n* = 14, SIG−1451: *n* = 16, unpaired *t*-test; *, ** *p* < 0.05, 0.01).

**Figure 3 ijms-23-08802-f003:**
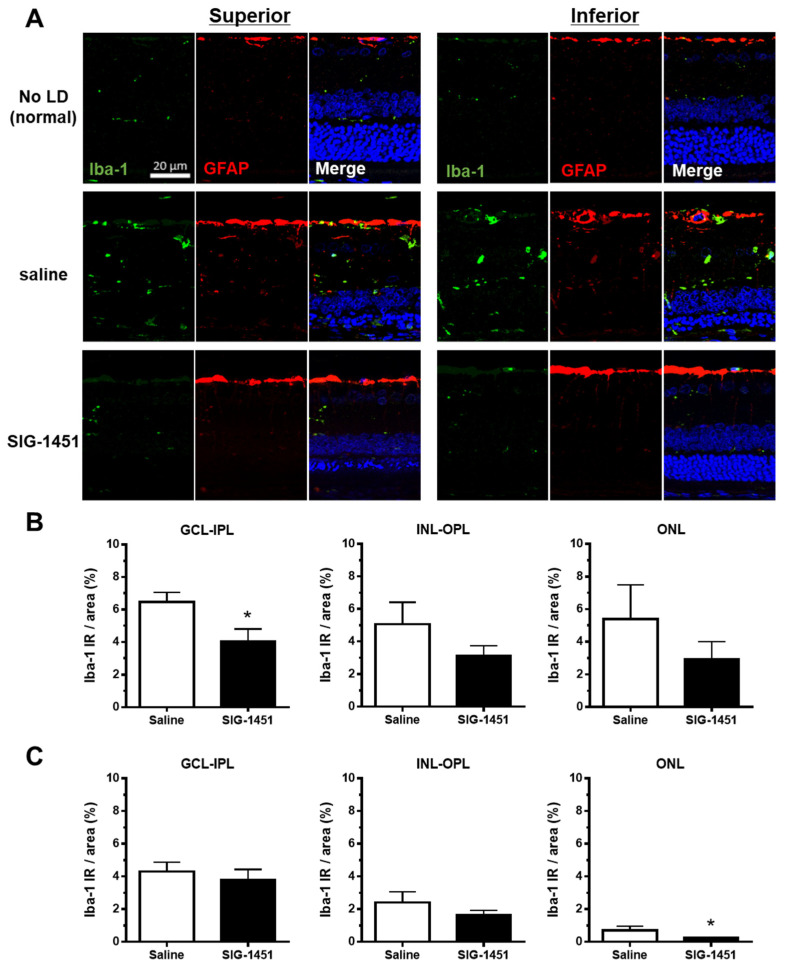
Iba-1 and GFAP immunohistochemistry in retinas from rats injected intraperitoneally with saline or SIG-1451. (**A**) Fluorescent microscopy images showing Iba-1 and GFAP immunoreactivity in the retina (**A**). Percentage of Iba-1 immunoreactive area to the total measured area of the superior (**B**) and inferior (**C**) parts of the retina. Data are represented in terms of the mean ± SE values (saline: *n* = 10, SIG-1451: *n* = 13, unpaired *t*-test; * *p* < 0.05). GCL: ganglion cell layer, IPL: inner plexiform layer, INL: inner nuclear layer, OPL: outer plexiform layer, ONL: outer nuclear layer.

**Figure 4 ijms-23-08802-f004:**
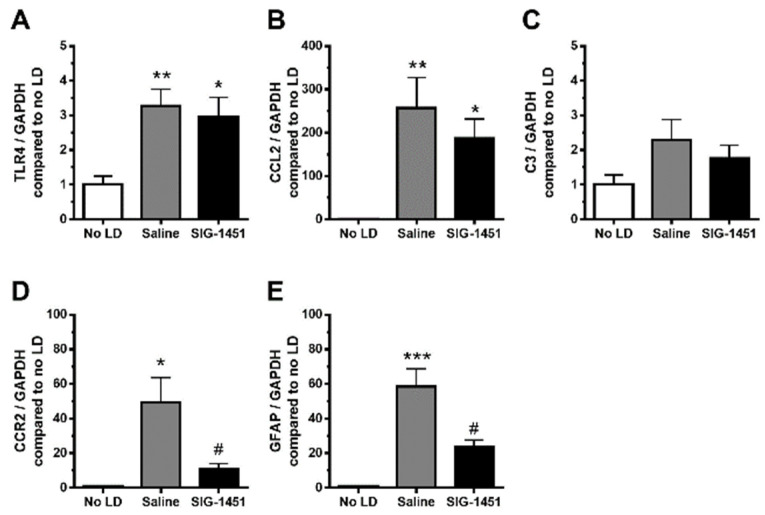
RT-qPCR analysis of inflammatory-response-related genes in light-damaged retinas. Gene expressions in the retinas in the light damage condition, compared to that of the internal control GAPDH, were normalised to those in the control condition. The expression levels of TLR4 (**A**), CCL2 (**B**), C3 (**C**), CCR2 (**D**), and GFAP (**E**) were quantified. Data are represented in terms of the mean ± SE values (saline: *n* = 8, SIG-1451: *n* = 8, Tukey’s multiple comparisons test; *, **, *** *p* < 0.05, 0.01, 0.001, ^#^ *p* < 0.05 compared to the saline group).

**Figure 5 ijms-23-08802-f005:**
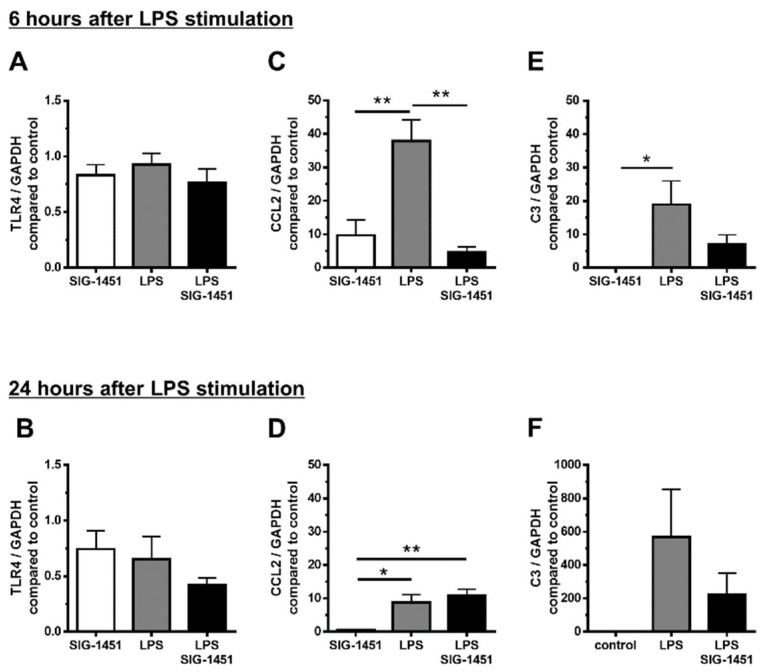
RT-qPCR analysis of inflammatory-response-related genes in cultured rMC-1 cells. Gene expression in rMC-1 cells was investigated for 6 h (**A**–**C**) and 24 h (**D**–**F**) after LPS stimulation. Independent experiments were performed. Data are represented in terms of the mean ± SE values (*n* = 6, Tukey’s multiple comparisons test; *, ** *p* < 0.05, 0.01).

**Figure 6 ijms-23-08802-f006:**
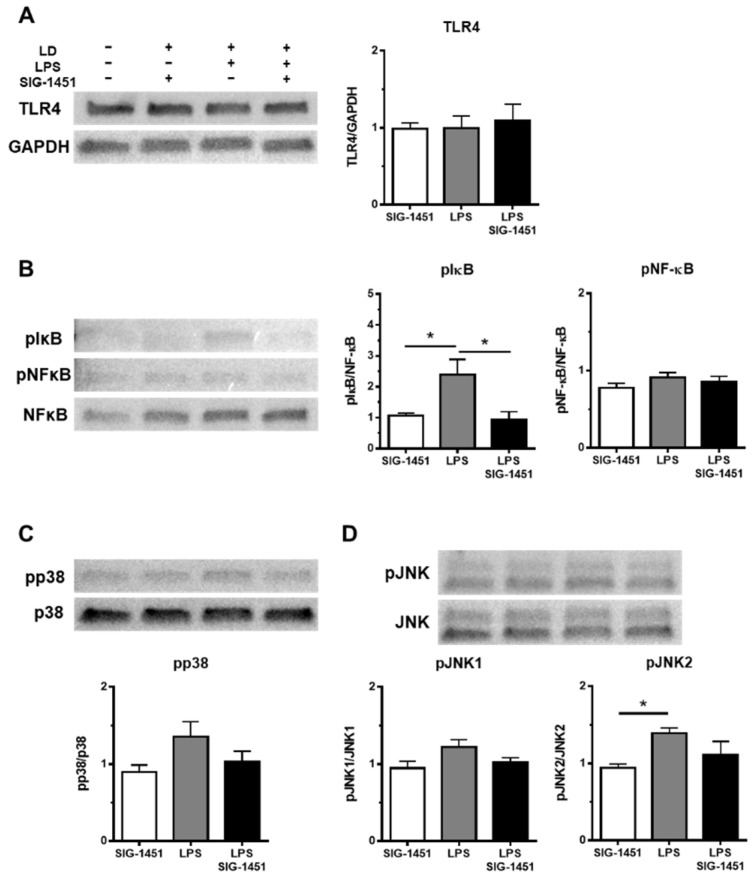
Western blot analysis of inflammatory response-related proteins in rMC-1 cells. Total cell lysates were prepared 3 h after the LPS stimulation and used for the Western blot analysis. The levels of TLR4 were not changed between groups (**A**). The phosphorylation of I-κB, but not NF-κB, was increased by LPS stimulation and blocked by the addition of SIG-1451 (**B**). Phosphorylated p38 levels tended to increase upon LPS stimulation (**C**). Phosphorylated JNK level, particularly pJNK2, was significantly increased by LPS stimulation (**D**). Data are represented in terms of the mean ± SE values (*n* = 5, Tukey’s multiple comparisons test; * *p* < 0.05).

**Table 1 ijms-23-08802-t001:** Primer pairs lists.

Name	Sequence (5′→3′)	Genbank
rat-GAPDH	F: AGGTCGGTGTGAACGGATTTG R:TGTAGACCATGTAGTTGAGGTCA	NM_017008.4
rat-Ccl2	F: CTGTCTCAGCCAGATGCAGTT R: GAGCTTGGTGACAAATACTACA	NM_031530.1
rat-Ccl12	F: TCGGAGGCTAAAGAGCTACA R: GTCCTTAACCCACTTCTCCTTG	NM_001105822.1
rat-Ccr2	F: ACACCCTGTTTCGCTGTAGG R: GTGCATGTCAACCACACAGT	NM_021866.1
rat-C3	F: ATCACGCCAAAGTCAAAGGC R: GTAGCATCCACGTCTCCCAA	NM_016994.2

**Table 2 ijms-23-08802-t002:** Antibodies lists.

ANTIGEN	SOURCE	DILUTION	MANUFACTURER
Iba1	rabbit	1/1000	Fijifilm Wako (013-27691)
GFAP	mouse	1/50	Cell Signaling (#3670)
NF-kB	mouse	1/1000	BD Biosciences (610868)
pNF-kB	rabbit	1/1000	Cell Signaling (#3031)
pIk-B	mouse	1/500	Santa Cruz Biotechnology (sc-8404)
JNK	rabbit	1/1000	Santa Cruz Biotechnology (sc-571)
pJNK	mouse	1/1000	Santa Cruz Biotechnology (sc-6254)
p38	rabbit	1/1000	Cell Signaling (#9212)
pp38	rabbit	1/1000	Cell Signaling (#9211)
TLR4	mouse	1/1000	Santa Cruz Biotechnology (sc-293072)
GAPDH	rabbit	1/1000	Santa Cruz Biotechnology (sc-25778)
Anti-rabbit IgG	goat	1/7500	Promega (S3731)
Anti-mouse IgG	goat	1/7500	Promega (S3721)

## Data Availability

The datasets used and/or analysed during the current study are available from the corresponding author upon reasonable request.

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
