# Peer review of "SIG-1451, a Novel, Non-Steroidal Anti-Inflammatory Compound, Attenuates Light-Induced Photoreceptor Degeneration by Affecting the Inflammatory Process"

_ijms, 2022, doi:10.3390/ijms23158802_

Round 1

Reviewer 1 Report

Kikuchi et al. utilised a light ablation rat model to assess the efficacy of SIG-1451 on maintaining photoreceptors and the underlying mechanism. The authors report that while SIG-1451 treatment did not maintain functional vision, that retinal thickness was partly retained for certain retinal regions in treated animals. Further, there was an increase in ERG response at 3 days post ablation in the treated animals. 

Overall, the manuscript is lacking in detail and description. I have many concerns about the way things are described, the work, and data presentation. Controls and description of methods are missing in several places. Further, there are experiments are inadequate and do not provide enough evidence for the conclusions. 

1. Introduction

- "it takes time to progress to dry AMD" -- I am unsure what you mean by this. Do you mean that onset is variable? There are things that increase a person's risk of developing AMD, such as smoking, a family history of AMD, or development of drusen or subretinal drusenoid deposits, but it is not a state that is generally described as 'progressed to'. Unusual language that does not line up with the AMD literature. Similarly, it is inaccurate to say that dry AMD "often transitions" to wet AMD. It certainly can progress, but only ~10% of AMD cases are wet AMD. This entire statement needs to be re-worked and appropriate citations added. 

- "some reports have shown that excessive inflammation accelerates retinal degeneration" only has one citation. This would be a single report. Either include more citations to change to say "a report has shown..."

- In the abstract you say that decreased phagocytic activity is associated with AMD but this is not mentioned in the Introduction. This needs to be discussed and the Abstract's language should be clarified. Specifically, RPE phagocytosis is decreased in AMD. How microglial and Müller cell phagocytosis are affected is unclear. 

2.1 -- Minor point: Section heading: the 'O' in 'optical coherence tomography' should not be capitalised. ERG had not been defined in the introduction/main text prior to this legend, and so "electroretinogram" or "electroretinography" should be written in full in the heading then defined, similar to OCT. 

- "retinal layer" is a confusing term. Saying 'whole retinal thickness' would be more clear and in line with the literature. Generally when someone says a "retinal layer" they are referring to a specific layer, such as the inner nuclear layer, for example, whereas the authors are looking at the entire retina and not a specific layer. I suggest changing the language. 

2.2 Figure 2 should have inclusion of non-light ablated control animals to give the reader context for the SIG-1451's impact on retinal thickness. While there is a statistically significant increase at some retinal locations, how does that compare to an animal that has not had retinal lesions? Is that small increase expected to be biologically significant? Additionally, the thickness measurements should include both positive and negative SE for both groups. Currently, it looks as though positive bars are only included for the SIG treatment group and negative bars only included for the saline group. This can trick the reader's eye. If the graph becomes too messy due to this inclusion, I would suggest changing the colours so that it is more clear and/or making the graph larger.

2.3 

- "normal retinas" = control/non-ablated retinas? 

- Figure 3 should be larger on the page. It is challenging to see the immunofluorescence. Further, microglia need to generally be stained, rather than just activated microglia by Iba-1. There is a possibility that the drug is toxic to microglia (which is not assessed by the authors), and therefore there could simply be fewer microglia in the SIG-1451 retinas, rather than that they are not being activated. This is a major issue that must be addressed, and undermines all of the author's conclusions. 

2.4 

- Figure 4 + related text: How was expression assessed? qPCR? Western blot? Microarray? ELISA? It is unclear in the main text and figure legend and must be clearly stated. Based on the methods section, it is qPCR, but that needs to be said. 

2.5 + 2.6

- What are rMC-1 cells? Describe in the text. Further, why were Müller glial cells selected for this assay, rather than immortalised or cultured microglia cells? I am unsure about the biological relevance of this choice and it is not justified anywhere. Immunofluorescence looking at Müller glia in the rat retinas should further help tie this in. 

- These experiments should be replicated with microglia cells, especially given the title of the paper. There is not enough evidence to support the conclusion that microglia cells are being affected by the drug treatment. 

- The figure legends are too vague and Figure 6's legend does not state the cell type being investigated. 

- Again, "investigated the expression of genes" -- how was this done? 

- "phosphorylation of NF-κB and I-κB 176 plays central roles in immune system functioning" is much too vague. Clarify. 

3. Discussion 

- It should be noted that microglial translocation to the subretinal space has been found protective to the RPE in rodent models of retinal degeneration. This should be considered by the authors and discussed. The authors also do not mention why there seems to be a spatial impact. 

Overall, there is insufficient evidence that preventing microglial activation is the means though which this drug works and the mild (potential) retinal neuroprotection should be further interrogated and compared with non-light treated controls. Yes, there was less Iba-1 staining in the SIG-1451 treated retinas, but it was not assessed whether this was due to microglial toxicity. There could have been less 1ba-1 staining because there were fewer microglia, rather than the microglia were not being activated. This is a major oversight that significantly weakens the paper and undermines its conclusions. This should be addressed via rigorous immunofluorescence and cell culture work -- specifically, the Müller glia cell work should be repeated with a microglial line. Ideally, there would be immunofluorescent assessment of Müller glia in the rat retina as well to tie in the rMC-1 cell work. 

Author Response

Thank you for your detailed review and giving us suggestions.

A kind through explanation is very helpful for us to revise our manuscript.

I revised our manuscript following reviewer’s comments.

I appreciate it if you re-review our manuscript.

1.Introduction

- "it takes time to progress to dry AMD" -- I am unsure what you mean by this. Do you mean that onset is variable? There are things that increase a person's risk of developing AMD, such as smoking, a family history of AMD, or development of drusen or subretinal drusenoid deposits, but it is not a state that is generally described as 'progressed to'. Unusual language that does not line up with the AMD literature. Similarly, it is inaccurate to say that dry AMD "often transitions" to wet AMD. It certainly can progress, but only ~10% of AMD cases are wet AMD. This entire statement needs to be re-worked and appropriate citations added. 

RE: The sentence was deleted because the IJMS is an international journal though the percentage of the dry AMD is about 10 % in Japan.

- "some reports have shown that excessive inflammation accelerates retinal degeneration" only has one citation. This would be a single report. Either include more citations to change to say "a report has shown..."

RE: Added the citation. (line 66)

[25] Elsherbiny, N. M.; Sharma, I.; Kira, D.; Alhusban, S.; Samra, Y. A.; Jadeja, R.; Martin, P.; Al-Shabrawey, M.; Tawfik, A., Homocysteine Induces Inflammation in Retina and Brain. Biomolecules 2020, 10, (3).

[26] Ma, W.; Zhao, L.; Fontainhas, A. M.; Fariss, R. N.; Wong, W. T., Microglia in the mouse retina alter the structure and function of retinal pigmented epithelial cells: a potential cellular interaction relevant to AMD. PLoS One 2009, 4, (11), e7945.

- In the abstract you say that decreased phagocytic activity is associated with AMD but this is not mentioned in the Introduction. This needs to be discussed and the Abstract's language should be clarified. Specifically, RPE phagocytosis is decreased in AMD. How microglial and Müller cell phagocytosis are affected is unclear. 

RE: “Decreased phagocytic activity” was deleted

2.1 -- Minor point: Section heading: the 'O' in 'optical coherence tomography' should not be capitalised. ERG had not been defined in the introduction/main text prior to this legend, and so "electroretinogram" or "electroretinography" should be written in full in the heading then defined, similar to OCT. 

RE: Title was changed followed the reviewer 3 suggestion. (line 84-85)

2.1. SIG-1451 rescues retinal thickness and improves electroretinographic responses in light induced photoreceptor degeneration

- "retinal layer" is a confusing term. Saying 'whole retinal thickness' would be more clear and in line with the literature. Generally when someone says a "retinal layer" they are referring to a specific layer, such as the inner nuclear layer, for example, whereas the authors are looking at the entire retina and not a specific layer. I suggest changing the language. 

RE: Changed "retinal layer" to 'whole retinal thickness' (line 97)

2.2 Figure 2 should have inclusion of non-light ablated control animals to give the reader context for the SIG-1451's impact on retinal thickness. While there is a statistically significant increase at some retinal locations, how does that compare to an animal that has not had retinal lesions? Is that small increase expected to be biologically significant? Additionally, the thickness measurements should include both positive and negative SE for both groups. Currently, it looks as though positive bars are only included for the SIG treatment group and negative bars only included for the saline group. This can trick the reader's eye. If the graph becomes too messy due to this inclusion, I would suggest changing the colours so that it is more clear and/or making the graph larger.

RE: The no LD retina was added (fig. 2). The vertical retinal sections through the optic nerve were made and the measurements of retinal thickness were performed from the optic nerve as a zero point.

The both positive and negative SE, and colours were added.

2.3 

- "normal retinas" = control/non-ablated retinas? 

RE: "normal retinas" was changed to “no light-damaged retina”. (line 147)

- Figure 3 should be larger on the page. It is challenging to see the immunofluorescence. Further, microglia need to generally be stained, rather than just activated microglia by Iba-1. There is a possibility that the drug is toxic to microglia (which is not assessed by the authors), and therefore there could simply be fewer microglia in the SIG-1451 retinas, rather than that they are not being activated. This is a major issue that must be addressed, and undermines all of the author's conclusions. 

RE: Changed to the large figure. (Fig. 3)

We could not show the direct effect of SIG-1451 on microglia, and we have no data whether SIG-1451 has toxicity to microglia. However, we observed the partial protective effect on the photoreceptor degeneration and the decreased Iba-1 immunoreactivity. We speculate that a primary effect of SIG-1451 may be to inhibit secondary degeneration mediated by inflammation.

2.4 

- Figure 4 + related text: How was expression assessed? qPCR? Western blot? Microarray? ELISA? It is unclear in the main text and figure legend and must be clearly stated. Based on the methods section, it is qPCR, but that needs to be said. 

RE: The Fig.4 title was changed

RT-qPCR analysis of inflammatory response-related genes in light damaged retinas

2.5 + 2.6

- What are rMC-1 cells? Describe in the text. Further, why were Müller glial cells selected for this assay, rather than immortalised or cultured microglia cells? I am unsure about the biological relevance of this choice and it is not justified anywhere. Immunofluorescence looking at Müller glia in the rat retinas should further help tie this in. 

RE: The source of rMC-1 cells was written in “4.8. Cell culture” of the “4. Materials and Methods”.(line 450)

Added the sentences indicated below.

Line 310-315

It is well-known that Müller cells protect neurons via a release of neurotrophic factors, the secretion of antioxidants, and the uptake or degradation of the excitotoxins by responding the various types of injury and accelerate neuronal degeneration through a reactive Müller cell gliosis [34]. To focus on the role of Müller cells in the protective effect of SIG-1451, we used the immortalized rat Müller cell line, rMC-1 cells.

- These experiments should be replicated with microglia cells, especially given the title of the paper. There is not enough evidence to support the conclusion that microglia cells are being affected by the drug treatment. 

RE: We showed that SIG-1451 inhibited the activation of microglia (Fig. 3), but it is unclear whether the inhibition of microglial activation was related to the protective effect as you indicated. The time is too short to do the additional experiment using microglial cells. So, I slightly changed the title.

“SIG-1451, a novel, non-steroidal anti-inflammatory compound, inhibits microglial activation and attenuates light-induced photoreceptor degeneration”

- The figure legends are too vague and Figure 6's legend does not state the cell type being investigated. 

RE: The title and legend in each figure were modified

- Again, "investigated the expression of genes" -- how was this done? 

RE: The titles of Fig. 5 & 6 were modified.

- "phosphorylation of NF-κB and I-κB plays central roles in immune system functioning" is much too vague. Clarify. 

RE: Added a few sentences indicated below.

Line 243-246

NF-κB exists in the cytoplasm as an inactive form by binding to I-κB. Various stimuli such as LPS, inflammatory cytokines, and viruses trigger the activation of I-κB kinases, which induce the phosphorylation of I-κB. NF-κB translocates into the nucleus because NF-κB is not able to bind to the phosphorylated- I-κB.  

  1. Discussion 

- It should be noted that microglial translocation to the subretinal space has been found protective to the RPE in rodent models of retinal degeneration. This should be considered by the authors and discussed. The authors also do not mention why there seems to be a spatial impact. 

RE: Added the sentences indicated below.

Line 322-327

Ma W et al reported that gene expression of chemotactic cytokines and cell adhesion molecules in RPE cells was altered by co-culture of activated microglia and that mRNA ex-pression of CCL2 in the RPE cells significantly increased by co-culture with activated mi-croglia [30]. Retinal microglia translocate into the outer retina [35] and accumulate in the subretinal space [36] under conditions of advanced age and photoreceptor injury. RPE cells may be involved in the inflammatory responses. CCR2 has been reported to be ex-pressed in macrophages in an inherited retinal degeneration model [37]. These results in-dicated that SIG-1451 might inhibit CCR2 expression in other retinal cells, such as mac-rophages activated by LD and the RPE cells, but not in Muller cells in vivo.

Overall, there is insufficient evidence that preventing microglial activation is the means though which this drug works and the mild (potential) retinal neuroprotection should be further interrogated and compared with non-light treated controls. Yes, there was less Iba-1 staining in the SIG-1451 treated retinas, but it was not assessed whether this was due to microglial toxicity. There could have been less 1ba-1 staining because there were fewer microglia, rather than the microglia were not being activated. This is a major oversight that significantly weakens the paper and undermines its conclusions. This should be addressed via rigorous immunofluorescence and cell culture work -- specifically, the Müller glia cell work should be repeated with a microglial line. Ideally, there would be immunofluorescent assessment of Müller glia in the rat retina as well to tie in the rMC-1 cell work. 

RE: Added the GFAP immunohistochemistry into the Fig. 3. We could not get an antibody for the Müller glia because of the limited time. So, we used a GFAP antibody that recognize astrocytes and activated Müller glia for immunohistochemistry

Line 295-299

The role of microglia in photoreceptor degenerations is still controversial. Funatsu J et al [30] reported that the depletion of resident microglia exacerbated cone cell death in rd10 mice. On the other hand, some researchers have shown that the increase of inflammation mediated by activated microglia is responsible for the pathogenesis and progression of AMD [25, 26].

Reviewer 2 Report

Very well written manuscript. No changes/suggestions

Author Response

Thank you for your detailed review.

I revised our manuscript following reviewer’s comments.

I appreciate it if you re-review our manuscript.

Reviewer 3 Report

Kikuchi et al has reported the protective effect of SIG-1451 in light induced retinal degeneration. The overall premise of the paper sounds impressive in terms of future therapeutic potential but the following points must be addressed to improve the clarity in the manuscript.

1. All the titles for the results should be re-phrased as the existing titles sound more like methods. for example: result 1.1 Measurements of retinal thickness using Optical coherence tomography (OCT) imaging and 81 ERGs. This should be re-phrased such as SIG-1451 rescues retinal thickness and improves electroretinographic responses in light induced photoreceptor degeneration 

2. Result 2.1 should also have before light damage OCT image. Also, the OCT images are not high resolution and clear, not sure how authors would have segmented them clearly. Same for figure 2

3. Result 2.1  Did authors see significant spatial differences in the retina light induced retinal degeneration and/or rescue using SIG-1451? if so, why. Especially the drug being administered systemically

4.Why do authors choose a cell culture system to compare the modulation  inflammatory response with inflammation related to light induced retinal degeneration. Also, is the inflammation observed by light induced damage to the retina is a cause or consequence of retinal damage? If it is a consequence will authors claim that the drug will treat the consequence not the cause. In that case will this drug could be used to treat inflammatory response in other retinal degenerative diseases. Authors could have performed the western blots in the retinal lysates and check for the inflammatory markers that they have analyzed in rMC-1 cells.

5.Authors in the discussion should clearly explain the mechanism behind the rescue of LD of the retina with SIG-1451,Is there evidence that this does this drug crosses BBB. Is the drug found in the retina after IP injection. Does this drug rescue retinal cell death directly or by inflammation mediated cell death?

6. Why did the authors choose 2 time points for OCT and ERG(3 day post LD vs 1 day post LD)

Author Response

Thank you for your detailed review and giving us suggestions.

A kind through explanation is very helpful for us to revise our manuscript.

I revised our manuscript following reviewer’s comments.

I appreciate it if you re-review our manuscript.

1. All the titles for the results should be re-phrased as the existing titles sound more like methods. for example: result 1.1 Measurements of retinal thickness using Optical coherence tomography (OCT) imaging and 81 ERGs. This should be re-phrased such as SIG-1451 rescues retinal thickness and improves electroretinographic responses in light induced photoreceptor degeneration 

RE: The titles were rephrased.

2.1. SIG-1451 rescues retinal thickness and improves electroretinographic responses in light induced photoreceptor degeneration

2.2. SIG-1451 protects the photoreceptor cells from the light-induced photoreceptor degeneration

2.3. Decreased Iba-1-like immunoreactivities in SIG-1451-treated rats

2.4. Up-regulation of CCR2 after the photoreceptor degeneration was inhibited by SIG-1451.

2.5. RT-qPCR analysis of inflammatory response-related genes in cultured rMC-1 cells

  1. Result 2.1 should also have before light damage OCT image. Also, the OCT images are not high resolution and clear, not sure how authors would have segmented them clearly. Same for figure 2

RE: The images before LD were added. The arrows and bars corresponding to the ILM-RPE and the ONL including the outer segments were shown in each image in Fig. 2 and 3.

  1. Result 2.1  Did authors see significant spatial differences in the retina light induced retinal degeneration and/or rescue using SIG-1451? if so, why. Especially the drug being administered systemically

RE: We are planning to develop SIG-1451 as an oral medicine.

4.Why do authors choose a cell culture system to compare the modulation inflammatory response with inflammation related to light induced retinal degeneration.

RE: Added the sentences indicated below.

Line 310-315

It is well-known that Müller cells protect neurons via a release of neurotrophic factors, the secretion of antioxidants, and the uptake or degradation of the excitotoxins by responding the various types of injury and accelerate neuronal degeneration through a reactive Müller cell gliosis [34]. To focus on the role of Müller cells in the protective effect of SIG-1451, we used the immortalized rat Müller cell line, rMC-1 cells.

Also, is the inflammation observed by light induced damage to the retina is a cause or consequence of retinal damage? If it is a consequence will authors claim that the drug will treat the consequence not the cause. In that case will this drug could be used to treat inflammatory response in other retinal degenerative diseases.

RE: In speculation, the inflammation is not a cause of photoreceptor degeneration, however, the inflammation is possibly related to the progression of photoreceptor degeneration. We think that SIG-1451 has the protective effect on the other retinal diseases with inflammation.

Added the sentences indicated below

Line 295-299

The role of microglia in photoreceptor degenerations is still controversial. Funatsu J et al [30] reported that the depletion of resident microglia exacerbated cone cell death in rd10 mice. On the other hand, some researchers have shown that the increase of inflammation mediated by activated microglia is responsible for the pathogenesis and progression of AMD [25, 26].

 Authors could have performed the western blots in the retinal lysates and check for the inflammatory markers that they have analyzed in rMC-1 cells.

RE: We haven’t done the western blot analysis in rat retinas. It is difficult to prepare the samples for the western blot because of a short revision time.

5.Authors in the discussion should clearly explain the mechanism behind the rescue of LD of the retina with SIG-1451, Is there evidence that this does this drug crosses BBB. Is the drug found in the retina after IP injection. Does this drug rescue retinal cell death directly or by inflammation mediated cell death?

RE: We have no idea whether SIG-1451 can pass through the BBB. In the revised version, GFAP immunohistochemistry was added. The fact that RT-qPCR were down-regulated in SIG-1451 treated retinas indicates that SIG-1451 directly or indirectly affected the retinal cells.

  1. Why did the authors choose 2 time points for OCT and ERG(3 day post LD vs 1 day post LD)

RE: In our previous studies, apoptosis of photoreceptor cells was seen 1 day post LD (ref. 17). We tried to check the histological and function changes of the early phase in photoreceptor degeneration.

Round 2

Reviewer 1 Report

The reviewer suggestions have not be adequately addressed and the manuscript has not been substantially improved. While some effort was put in to improving the manuscript, there has not been enough progress. There is inadequate background given in the Introduction, major conclusions as they relate to microglia are not supported, and logic behind experimental choices have not been given. Major points:

- Some of the Introduction sections that were confusing or inaccurate (such as the descriptions of dry vs wet AMD) were removed instead of corrected, which makes the Introduction thin and hard to follow. It simply does not provide the non-expert reader with enough information. Why mention dry and wet AMD at all if they are not explained? Why should the reader care about these conditions? What are they? There is also no mention of the genetic component of AMD nor hallmark features of the condition. Wholly inadequate, unfortunately. 

- The Introduction citations only refer to rodent and small mammal models of photoreceptor degeneration. Again, inadequate. There are many others, including zebrafish (which are becoming increasingly popular and have many genetic and inducible models of retinal disease), pigs, dogs, Xenopus, et cetera. There are reviews on these, as well. If the authors are specifically focussing on rodent/small mammal models intentionally, this should be stated and why this is so explained in detail. There is little to no explanation as to why rats were selected as the model for this research, either. 

- The retinal measurement graphs look much better now. The colours and +/- SE bars make them much more informative. However, the figure legend is confusing in terms of the p-values. I have never seen them written so unusually. It should be clear that * = p<0.05 and ** = p<0.01, which it is not now. It looks as though ** = p<0.01 and that * has no meaning. Furthermore, all abbreviations should be defined in the figure legend. Additionally, any images where measurements are taken should include scale bars, as without them it is difficult for the reader to assess the change in size or scale comparisons between the images. 

- "IR" is not defined. 

- There should be a no light damage group treated with SIG-1451. While yes, there are now images showing microglia, this does not address the question of whether SIG-1451 is affecting the microglia population/killing microglia. Microglia toxicity must be assessed. You need to compare vehicle control non-LD animals to SIG-1451 non-LD animals and quantify the microglia (either total # or area covered or something). Alternatively, it this is not possible, you could do rigorous toxicity assessments with cultured microglia, using a vehicle control group and SIG-1451 treated groups with different concentrations of the drug. Ideally both would be performed. Ultimately, microglia toxicity has not been adequately addressed, and as a consequence your results do not support the conclusions

- The added description of NF-kB and I-kB is appreciated, but still too vague. What is NF-kB? What kind of protein and what is its function? 

- You still not do explain what rMC cells are or why you chose them for your experiments. This is an important piece of logic and detail that is missing. 

- Again, the cell culture experiments should be repeated with microglia. You are claiming in your title and manuscript text that microglial activation is inhibited, but you have not shown that appropriately in retinal tissue or cell culture. 

Author Response

To reviewer 1,

Thank you for your detailed review and giving us suggestions.

A kind through explanation is very helpful for us to revise our manuscript.

I revised our manuscript following reviewer’s comments.

I appreciate it if you re-review our manuscript.

Reviewer 3 Report

Authors have answered most of the comments therefore I recommend this article for publication.

Author Response

To reviewer 3,

Thank you for your detailed review and for giving us suggestions.

A kind explanation is very helpful for us to revise our manuscript.

I believe our manuscript includes some helpful information for AMD researchers.

Thank you again.